# Constructing a Visual Dataset to Study the Effects of Spatial Apartheid in South Africa

**Raesetje Sefala**[1,2]     **Timnit Gebru**[1]     **Luzango Mfupe**[3]     **Nyalleng Moorosi**[4]

**Richard Klein**[2]

[1]The Distributed AI Research Institute, Palo Alto, California, USA
[2]School of Computer Science and Applied Mathematics
University of the Witwatersrand, Johannesburg, South Africa
[3]Council of Scientific & Industrial Research, NGEI, Scientia Campu, Pretoria, South Africa
[4]Google Research, South Africa

## Abstract

Aerial images of neighborhoods in South Africa show the clear legacy of apartheid, a former policy of political and economic discrimination against non-European groups, with completely segregated neighborhoods of townships next to gated wealthy areas. This paper introduces the first publicly available dataset to study the evolution of spatial apartheid, using $6,768$ high resolution satellite images of $9$ provinces in South Africa, $550$ of which are labeled. Our dataset was created using polygons demarcating land use, geographically labelled coordinates of buildings in South Africa, and high resolution satellite imagery covering the country from 2006-2017. We describe our iterative process to create this dataset over two years, which includes pixel-wise labels for $4$ classes of neighborhoods: wealthy areas, non wealthy areas, nonresidential neighborhoods and background (land without buildings). While datasets 7 times smaller than ours have cost over $\$1M$ to annotate, our dataset was created with highly constrained resources. We finally show examples of applications examining the evolution of neighborhoods in South Africa using our dataset.

## 1 Introduction

Analyzing many time-lapse satellite images presents the opportunity to study cities using computer vision, and create tools that allow governments and other entities to examine the effects of various policies. These tools can be used to further marginalize already disenfranchised communities by surveilling them and limiting their access to capital and opportunities [66, 3, 34, 17, 4, 5, 57, 33]. They can also be used to examine the effects of discriminatory policies [16, 47, 63, 46, 62, 43, 15], a line of study which we hope to support with this work. We present a satellite imagery dataset that can be used to analyze the effects of spa-

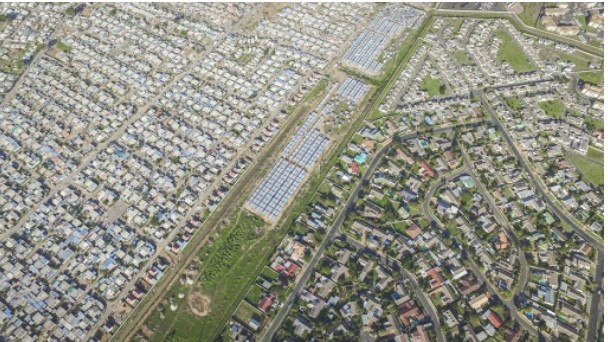

Figure 1: An example of spatial apartheid's legacy in Cape Town, South Africa, showing segregated neighborhoods of townships next to wealthy neighborhoods [42].

35th Conference on Neural Information Processing Systems (NeurIPS 2021) Track on Datasets and Benchmarks.

tial apartheid in South Africa. As noted in the associated datasheet, to guard against negative use cases of such a dataset, the dataset will only be available upon request for approved research purposes [50].

Apartheid is a former South African policy of segregation, and political and economic discrimination against non-European groups in the country [64]. The Group Areas Act passed in 1950 [9] forcefully relocated Black, Coloured and Indian people out of urban areas and into townships where they were allocated uniformly sized small plots of land on the outskirts of cities and towns. While apartheid legislation was repealed in 1991, its effects are still alive today [45]. For example, Figure 1 shows aerial images taken by photographer Johnny Miller, depicting completely segregated neighborhoods of townships next to gated wealthy neighborhoods that have largely remained unaffected by the end of apartheid [42]. Although this effect is immediately obvious to any human looking at the photos, people cannot analyze large numbers of such images to gain insights.

We present the first publicly available satellite imagery dataset of South Africa that is suitable for the study of spatial apartheid, with pixel level annotations of 4 neighborhood classes: wealthy neighborhoods, non wealthy neighborhoods, nonresidential areas and background, in the 9 South African provinces. As seen in Figure 1, townships and wealthy residential neighborhoods can have distinct visual characteristics: e.g. the latter are usually more sparsely populated and green, while townships are densely populated but with a grid-like structure. These visual differences allow us to train a model distinguishing between different types of neighborhoods. Although the majority of works in computer vision focus on algorithmic development, the most critical and time-consuming step in projects such as this is procuring and labeling the necessary datasets for the task [32]. Over two years, we performed an iterative process that uses the insights gained from baseline models to understand the shortcomings of our dataset, adding new elements to the data as necessary. The final dataset consists of geo-referenced satellite images covering the entire country of South Africa and a corresponding mask of neighborhoods labeled according to their type. This mask was built from a combination of geo-referenced polygons called Enumerator Areas (EAs) subdividing land-use as specified by the government, and data points locating all buildings in South Africa. To our knowledge, this is the first dataset for land cover classification covering an entire African country, that is suitable for the study of spatial apartheid or related phenomena.

The rest of this paper is structured as follows. Section 2 discusses related work, Section 3 introduces the components used to create our visual dataset (satellite images, land use polygons, and building points). We describe the methodology used to construct our dataset in Section 4, and discuss various challenges unique to this task. We present experiments to further understand our dataset in Section 5, and show preliminary results estimating the evolution of neighborhoods using our dataset. We conclude by discussing broader impacts in section 6.

## 2   Background and Related Work

While there are several freely available datasets for the broader task of land cover classification [10, 65, 14, 58, 25, 60, 2, 38, 61], most of them have been created for the developed world [53, 20, 60, 2, 38, 61], and a handful using images from developing countries which have very different characteristics from neighborhoods in South Africa [30, 1, 59, 48]. Furthermore, some of these datasets have different objectives from ours, such as classifying buildings, cars, trees, sidewalks, bodies of water and urban areas, without distinguishing between different types of neighborhoods [60, 2, 38, 61].

In addition, datasets like [10] do not differentiate between neighborhood types within the urban land class–a task essential to studying spatial apartheid. The UC Merced dataset labels neighborhoods ranging from dense to sparse residential as part of 18 other classes [65]. Although these classes are more detailed and more closely resemble our task, they cover cities in the United States of America which do not share similar visual characteristics with those in South Africa [30].

On the other hand, publicly available datasets denoting land-use in the African continent usually only have 2 classes distinguishing between informal settlements and everything else, and typically only cover a single city [39, 23, 51] due to cost constraints. Outside of Africa, it cost the Chesapeake Conservancy over 10 months and 1.3 million dollars to create a six-class land cover dataset covering the Chesapeake Bay watershed [53, 20], an area $\sim 7$ times smaller than the size of South Africa.

Although there are some relevant datasets covering small sections of South Africa [6, 24, 41, 28], they are either not publicly available, are very outdated, do not have labels which would allow us to study

neighborhood types at a higher granularity than formal versus informal settlements [28, 12, 37, 40, 19], or address entirely separate tasks from ours such as building detection [7]. Furthermore, in a resource constrained setting such as ours, the methodologies used to create these small datasets cannot be used to cover the entire country. While we found one dataset for land cover classification that covers all of South Africa [11], this dataset was created using proprietary data (rather than publicly available ones such as ours). Most importantly, it does not distinguish between wealthy and non wealthy neighborhoods, a distinction crucial for the study of spatial apartheid.

## 3 A Neighborhood Segmentation Dataset

| Year | Resolution | Number of images |
|------|------------|------------------|
| 2006 | 7,550 x 7,250 | 625 |
| 2007 | 7,550 x 7,250 | 690 |
| 2008 | 21,688 x 21,688 | 545 |
| 2009 | 29,406 x 29,277 | 545 |
| 2010 | 21,688 x 21,688 | 545 |
| 2011 | 21,688 x 21,688 | 550 |
| 2012 | 21,688 x 21,688 | 531 |
| 2013 | 21,688 x 21,688 | 549 |
| 2014 | 35,000 x 35,000 | 549 |
| 2015 | 35,000 x 35,000 | 548 |
| 2016 | 35,000 x 35,000 | 548 |
| 2017 | 35,000 x 35,000 | 543 |

Table 1: Satellite Images in our dataset. We create labels for 2011 images.

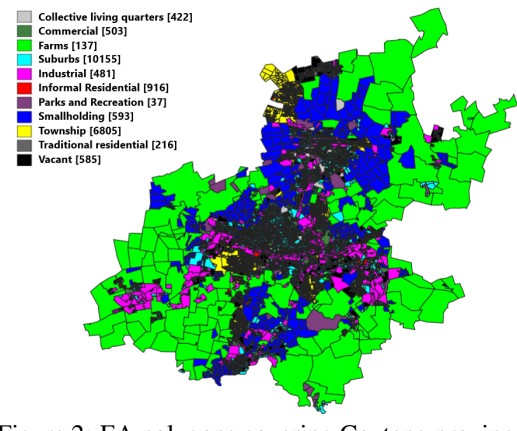

Figure 2: EA polygons covering Gauteng province after distinguishing between townships and suburbs comprising the formal residential class.

**Satellite images:** We obtained satellite images covering the entire country of South Africa from 2006-2017 from the South African National Space Agency [56]. The dataset consists of images taken from the SPOT sensor, with varying resolutions in different years as depicted in Table 1. Given that our ground truth labels were obtained in 2011, we also use satellite images from 2011 to assemble our labelled dataset. These images are at a resolution of 10m (each pixel represents 10 meters on the ground), and use the EPSG:4326 (WGS 84/Latlong) coordinate system. Each image is $21,688 \times 21,688$ pixels and the entire country is covered by a total of $550$ images.

**Enumeration Areas:** To associate image pixels with the types of neighborhoods they depict, we turned to the Enumeration Areas (EAs) dataset created in 2011 by Statistics South Africa [55]–a government agency responsible for conducting the census. The dataset consists of land demarcations according to 1 of 11 government-specified use cases (e.g. farms, industrial areas, residential areas, etc.). EAs are geographical units consisting of 100-250 households, used to demarcate locations for which census data is aggregated. One shortcoming of the EA dataset is that townships are grouped with suburbs under the label "formal residential areas", which does not allow us to distinguish between them. However, since suburbs are much wealthier than townships, it is crucial to distinguish between them in order to study spatial apartheid. In addition to allocating collective living quarters, villages and townships to non European residents, the Group Areas Act of 1950 allocated a much lower budget for these neighborhoods than others such as small holdings, suburbs and farms.

In order to identify townships that may be annotated as formal residential areas, we took the steps outlined below. The EA dataset consists of attributes representing data aggregated at different resolutions. EAs are the smallest unit of aggregation, followed by Sub-places, Main-places, Districts and continuing on to Provinces. We first obtained a list of all the Main-places that are annotated as formal residential areas, resulting in 1,655 Main-places consisting of a combination of suburbs and townships. Then, we recruited 10 graduate students at the University of Witwatersrand. Each student was born and raised in one of the 9 provinces, with 2 students from the KwaZulu-Natal province, the largest province in South Africa (by population size and area). The students' task was to check if any of the Main-places that were labeled as formal residential areas were indeed townships. Each of the 10 students went through all 1,655 Main-places and labeled those they believed to be townships as such, along with their level of certainty (certain or uncertain). For this procedure students often

asked their relatives or others living in the townships to verify their labels. If 3 or more students agreed that a Main-place is a township, we labeled that neighborhood as a township. To validate the labels, we further used additional sources such as Wikipedia and property websites such as `https://www.privateproperty.co.za/` and `https://www.property24.com/` to retrieve a list of townships and suburbs in South Africa respectively. We assembled a list of 362 townships from Wikipedia. If all 3 annotators agreed that a Main-place was a township but it was not listed on Wikipedia as such, we kept the township label. If one or more student listed a Main-place as a township, and Wikipedia (and other sources) listed it as such, we labeled it as a township. The most likely source of error from this step is mistakenly identifying townships as suburbs since labels are obtained for 2011 and students and their relatives have to remember which neighborhoods were townships in 2011 even if the neighborhoods have changed. In addition, not all townships are listed on Wikipedia. Figure 2 shows EAs covering the Gauteng province along with their associated land-use labels, after distinguishing between townships and suburbs in formal residential areas.

**Geo-referenced buildings dataset:** The Geo-referenced buildings dataset tells us the locations of all buildings in South Africa. This shapefile dataset was created by Eskom (a South African electricity public utility company) in partnership with the Council for Scientific and Industrial Research, and consists of building count data in South Africa from 2006 to 2016. The dataset captures geographical coordinates of formal, informal and non-dwelling structures per year over a period of 10 years. To annotate our dataset, we use the building count data from 2011 consisting of 12,515,847 buildings in South Africa, as the EA dataset is only available in 2011.

The only publicly available reports we have found for the dataset are from 2007 and 2010, which describe the labeling procedure and potential sources of error [13, 44]. As noted in [44], "All the mapping and classification of the structures are done through image interpretation and no field work has been conducted at this stage of the project." The initial dataset was created in 2006, and used as a basis for 2007, only updating buildings that changed in the last year (either new buildings or demolished buildings). Data for subsequent years was also created using the same procedure: using the dataset from the previous year as the base layer for the next year of interest. A random sample of the dataset was selected from across the country, and evaluated by independent labelers which corrected points with high false positives and false negatives. For instance, [13] notes that the highest false negative value in an urban sample was 1.34%, and the highest false positive rate was 1.20%. Rural areas have a higher error rate, with the highest false positive rate being 17.39%, and the highest false negative rate 2.26%. Some sources of error for this dataset include human labeling error, inaccurate counts of buildings which are close to each other in high density neighborhoods (e.g. informal settlements), and clouds on satellite images obstructing the view of buildings. Like our satellite images, the data points use the EPSG:4326 (WGS 84/Latlong) coordinate system.

## 4 Dataset Creation Methodology

In addition to manual inspection during the dataset construction process, we used a U-Net [54] based semantic segmentation model to help evaluate the quality of the data, assist in the creation of ground truth labels, and guide the search for supplementary sources of data. U-Net based architectures have won several semantic segmentation challenges and performed state-of-the-art on neighborhood classification tasks since its introduction in 2015 [26, 18, 1], and its efficiency allows us to train and evaluate models quickly. This was particularly important while trying to understand the nature of our dataset. As we discuss in the sections below, we constructed our dataset using an iterative process where we train a model that allows us to see shortcomings in our training data while examining the results using our validation data, then augment/alter our dataset as necessary, and repeat the process.

We modified [54]'s U-Net semantic segmentation architecture to accept input data sizes of $80 \times 80$, $256 \times 256$ and $2,711 \times 2,711$. We saw poor performance with $80 \times 80$ and $2,711 \times 2,711$ images. This is perhaps because the model sees a small region at a time in the former case, given the input image size and the resolution of the satellite images (10m per pixel), and a region that is too large in the latter. Further details of the hyperparameters used to train the U-Net are in Supplementary A.1.2.

**Evaluation metrics:** In cases such as ours with imbalanced datasets, the accuracy metric can be misleading. Thus, we also use the Cohen's Kappa metric [31] which measures how well our classifier performs relative to what is expected by chance, if labels were predicted using a random classifier:

$$\kappa = \frac{p_o - p_e}{1 - p_e} = 1 - \frac{1 - p_o}{1 - p_e} \tag{1}$$

| Class | Number of pixels | | | % of pixels | | |
|---|---|---|---|---|---|---|
| | Train set | Validation set | Test set | Train set | Validation set | Test set |
| Suburbs | 67,336,557 | 15,043,710 | 35,303,660 | 57.22 | 12.78 | 30.00 |
| Townships | 204,255,256 | 24,861,153 | 65,076,127 | 69.43 | 8.45 | 22.12 |
| Informal settlement | 102,382,989 | 9,833,663 | 379,950,086 | 68.16 | 6.55 | 25.29 |
| Village | 85,567,053 | 32,715,623 | 12,871,645 | 65.24 | 24.94 | 9.81 |
| Small holdings | 12,464,368 | 2,468,759 | 5,692,240 | 60.43 | 11.97 | 27.60 |
| Collective living quarters | 7,434,193 | 1,455,496 | 3,008,361 | 62.48 | 12.23 | 25.28 |
| Industrial land | 41,957,836 | 7,542,567 | 13,488,518 | 66.61 | 11.97 | 21.41 |
| Commercial land | 14,384,681 | 1,708,257 | 2,558,774 | 77.12 | 9.16 | 13.72 |
| Parks and recreation | 3,544,161 | 103,664 | 787,369 | 79.91 | 2.34 | 17.75 |
| Farms | 75,362,779 | 43,385,159 | 62,256,169 | 41.64 | 23.97 | 34.39 |
| Vacant | 4,575,672 | 1,050,783 | 1,608,610 | 63.24 | 14.52 | 22.23 |
| Background | 4,467,376,631 | 1,709,519,230 | 1,609,041,305 | 57.38 | 21.96 | 20.67 |

Table 2: Number of pixels per class for the subset of the data we iterated on during the construction of our dataset, labeled using the EA and building datasets.

Assume there are 2 classifiers: our classifier and a random classifier. $P_o$ is the empirical probability of agreement between the two classifiers, and $P_e$ is the expected probability of agreement, estimated by calculating the empirical agreement when both classifiers randomly assign labels. In highly imbalanced datasets, one can still achieve high accuracy while assigning the most common label to all classes, whereas this would be detected with the $\kappa$ value which would be close to zero.

**Data preparation:** We perform all spatial data processing tasks using the QGIS software [52]. To align all components of our dataset, we re-project datasets using different geographic coordinate reference systems to EPSG:4326 (the system used by our satellite images). This is important since accurate building masks can only be obtained if the datasets can be accurately overlaid.

While constructing and refining the dataset, we iterated on 19 satellite images from Gauteng, Limpopo, North West, Free State and Mpumalanga provinces (details in Supplementary A.1.1). This process involved training and testing a model and interpreting what kind of data should be added in order to create better labels in the next iteration. To do this, we split the data into a $60 : 20 : 20$ training, validation, test set ratio, ensuring that the same pixel does not appear in more than one set.

We then divided images such that classes are well balanced according to the split ratio. To do this, we counted the number of pixels per class per image, and added each image to a particular split by attempting to have a $60 : 20 : 20$ ratio for each class. This was done by performing a grid search for a $60 : 20 : 20$ percentage of pixels per class split over our images. Out of the 19 images, 11 are in the training set, 4 in the validation set and 4 in the test set. We tile the 19 images of size $21,688 \times 21,688$ pixels into 134,064 images of size $256 \times 256$. We further balanced the training data by discarding images with $30\%$ or more vacant land from the split (details in Supplementary A.1.2). Since our final dataset has 550 satellite images, we only used $3.45\%$ of our dataset in the tuning process.

Although we strove to have close to a $60 : 20 : 20$ split for each class, as shown in Table 2, it was not possible to have that ratio for all classes. In particular, for the "Parks and Recreation" class we have an $80 : 2 : 18$ split which was not our goal. For classes such as "Background" however, we were able to obtain close to the desired split. While images in each split are unique to that set, part of a city/neighborhood in one set can exist in the other if they share a boundary.

**Using solely the EA dataset as Ground-truth:** As a first step, we used the polygons from the EA dataset described in Section 3 to create image masks. The subset of our data used for iteration contains 27,636 polygons (a subset of the 103,576 EAs covering South Africa). To create image masks, we read each polygon, spatially located its corresponding group of pixel coordinates on the satellite image, and labeled the corresponding pixels according to a specified key set on a blank image associated with each satellite image. The images and corresponding masks are in the lossless PNG format as we need to preserve the exact pixel values for the masks.

If one polygon spans two or more images, this algorithm looks for pixels corresponding to the polygon on each image it spans. One disadvantage of this process is that each polygon iterates every satellite image in the dataset. To make this process more efficient, we used the QGIS [52] software to dissolve the polygons into our 12 classes (Figure 3(c)). For example, we merged all 6,380 polygons representing township neighborhoods in our dataset into one large polygon. This significantly reduces the amount of time the CPU spends reading polygons.

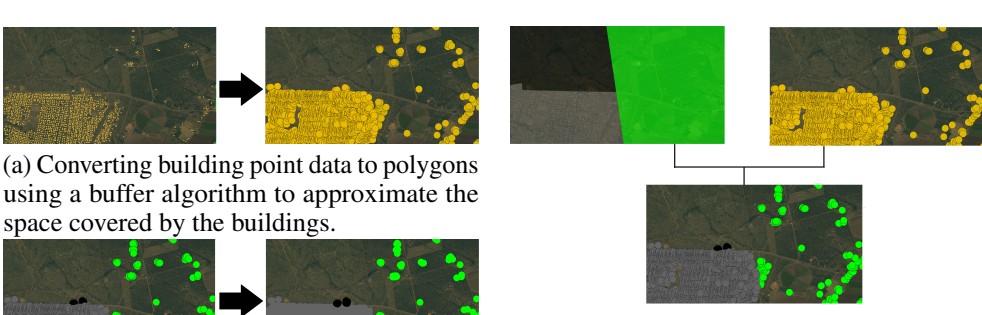

(a) Converting building point data to polygons using a buffer algorithm to approximate the space covered by the buildings.

(c) Dissolving overlapping building polygons by neighborhood. Images are of size 786 x 386 pixels and resolution 2.5m per pixel.

(b) Computing the spatial intersection between the land use labels from the EA dataset and the buffed building polygons so that we can label the types of neighborhoods these houses are in.

Figure 3: Data processing steps.

We trained a U-Net model on this dataset (details in Supplementary A.1.2), and examining the results uncovered some shortcomings of the dataset. Table 3 shows that while pixels in the dataset were classified with 61% accuracy, as shown by the low Cohen's Kappa value (0.0151), this is due to the over-representation of farmland (57% of pixels in the training data). Further, since the EA labels only specify the designated land use, vacant, farm, commercial and industrial lands can be confused for each other because undeveloped land often looks like vacant land (see Supplementary A.1.3). A clear next step from this iteration was to create ground truth labels which enable us to distinguish between background pixels and those representing buildings.

**Using the EA and building datasets as Ground-truth:** In order to label our satellite images with building locations, we used the geo-referenced buildings dataset described in Section 3, and followed the procedure below to create masks assigning each pixel in a satellite image to the desired class.

- Buffing points into polygons: The first step was to use the buffer algorithm to transform each point into a circular polygon of a specific radius. In our case, we inflated the points by a distance of 0.0007 decimal degrees. We arrived at this number through a trial and error search, looking for polygons which covered an average suburban house and its yard (Figure 3(a)). Buffing allows large swaths of vacant land to be labelled as background. See Supplementary A.1.4 for details.

- Spatial Intersection: To label which neighborhood each building belongs to, we computed the spatial intersection of the EA and buffed building datasets as demonstrated in Figure 3(b), joining the datasets at their overlapping points.

- Dissolve polygons by neighborhood types: Before this step, the polygons consisted of over 12 million data points each saved in shapefiles, resulting in computationally expensive read and write operations to convert them to image masks. Dissolving significantly reduced the number of polygons by grouping those that belong to the same neighborhood together.

- Create masks: We overlaid the dissolved polygons on the satellite imagery to create labels designating each pixel as one of the 11 classes or background if it does not denote a building.

Training a U-Net model on this dataset achieves an accuracy of 92.94% and a Cohen's Kappa value of 0.6299 on the 12 classes, a significant improvement from the previous value of 0.0151 (results summarized in Table 3). Since the building polygon dataset was created using building data points, labels on farmland, for example, depict the buildings on farms and not the farm itself. Further analyzing the results, we saw that most of the confusion was between classes with similar visual characteristics like farm houses and smallholdings, and it is unlikely that even a human would be able to distinguish between these classes visually. To alleviate this confusion, we collapsed the 12 classes in our dataset into 4 visually distinct categories, combining classes with similar visual characteristics: background (all land without buildings), wealthy neighborhoods (suburbs, smallholdings, and farms), non wealthy neighborhoods (townships, informal settlements, villages and collective living quarters), and nonresidential building clusters (commercial areas, industrial areas, buildings on vacant land, parks, and recreational areas). The wealthy/non wealthy neighborhood demarcation was also informed by examining real estate prices on websites such as www.property24.co.za. Small holdings,

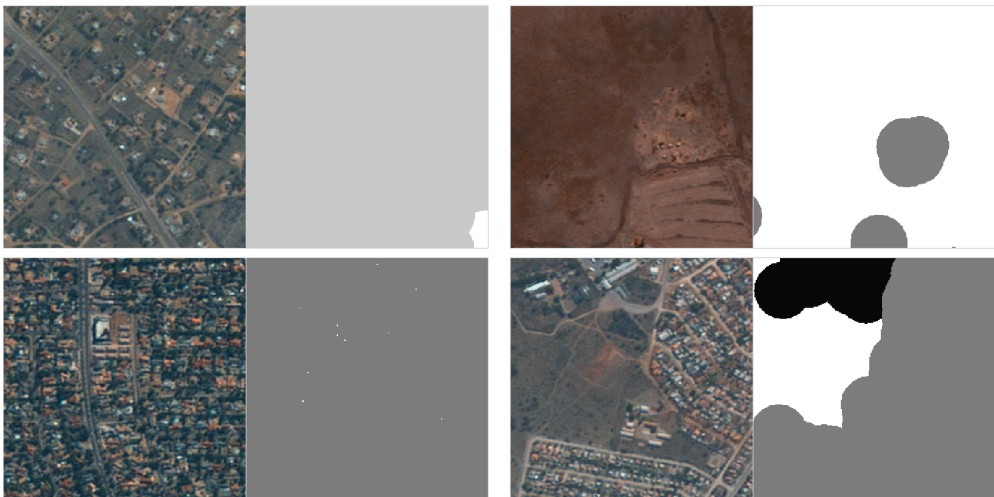

Figure 4: Samples of image mask pairs from our dataset. White: background, black: nonresidential neighborhood, light gray: non wealthy neighborhood, dark gray: wealthy neighborhood.

| Dataset | Accuracy | Cohen's $\kappa$ |
|---|---|---|
| EA data: 12 classes | 61% | 0.0151 |
| EA data + buildings: 12 classes | 92.94% | 0.6299 |
| EA data + buildings: 4 classes | 96.14% | 0.7578 |

Table 3: Classification accuracy for various ground truth modifications. "EA" is an abbreviation for the Enumeration Area dataset.

| Class | Pixels (#) | Pixels (%) |
|---|---|---|
| NW | 2,787,947,606 | 1.07 |
| W | 3,081,228,682 | 1.18 |
| NR | 360,768,002 | 0.14 |
| B | 254,193,735,710 | 97.61 |

Table 4: The number and percentage of pixels per class for the final dataset with classes NW=non wealthy, W=wealthy, NR=nonresidential, B=background.

suburbs and farms are much more expensive than villages, townships and collective living quarters (informal settlements do not even appear on these websites for sale). However, the price of some collective living quarters near locations with high economic development can be closer to wealthy neighborhoods. This can be a source of error in our demarcation of wealthy and non wealthy neighborhoods. Furthermore, while townships were allocated low budgets during apartheid, there are now wealthy households in townships post apartheid. The majority of households, however, are still non wealthy. If an entire township or other neighborhood becomes wealthy, its visual characteristics also change to look more like the neighborhood types we have classified as such. Table 3 shows that performance on the combined classes is at 96.14% accuracy and 0.75 Cohen's Kappa.

**Final dataset composition:** Our final labeled dataset consists of 550 satellite images and masks from 2011, upsampled to a resolution of $21,760 \times 21,760$ (details in A.2.1). Figure 4 shows a few samples from our dataset and Table 4 summarizes the class distribution (more examples in A.1.5).

## 5   Experiments

Here, we first perform experiments on a subset of our dataset (1,869,840 images) to understand its characteristics, and then provide examples of the types of analyses we hope it can be used for. Our supplementary materials (A.2.1-A.2.2) provide details on how we sampled this data and created the training, validation and test splits (Table 6). Since our dataset consists of mostly vacant land, we created a subset which covers a variety of sceneries such as densely/sparsely populated areas, mainland/coastal land and dif-

| Split | Satellite | Tiled |
|---|---|---|
| Training set | 60 | 1,121,904 |
| Validation set | 20 | 373,968 |
| Testing set | 20 | 373,968 |
| Total | 100 | 1,869,840 |

Table 5: The number of images in each split for our baseline experiments. Satellite refers to the $21,760 \times 21,760$ resolution satellite images and tiled refers to the $256 \times 256$ images.

ferent ecosystems such as forests and grassland.

| Neighborhood | U-Net [54] | | | | | DeepLabV3+ [8] | | | | |
|---|---|---|---|---|---|---|---|---|---|---|
| | 99% | 85% | 75% | 65% | 50% | 99% | 85% | 75% | 65% | 50% |
| Wealthy | 0.997 | 0.966 | 0.946 | 0.927 | 0.895 | 0.997 | 0.952 | 0.936 | 0.926 | 0.907 |
| Non Wealthy | 0.997 | 0.966 | 0.948 | 0.933 | 0.915 | 0.998 | 0.955 | 0.940 | 0.927 | 0.911 |
| Nonresidential | 0.995 | 0.903 | 0.863 | 0.837 | 0.809 | 0.995 | 0.910 | 0.884 | 0.868 | 0.854 |
| Background | 0.999 | 0.992 | 0.988 | 0.987 | 0.985 | 0.999 | 0.988 | 0.988 | 0.989 | 0.989 |

Table 6: Precision at 50% to 99% intersection over union (IOU) per class: i.e., the precision for each class, taking only segmentations of neighborhoods achieving an IOU of 50% or more, 65% or more, 75% or more, 85% or more, and 99% or more in each image as true positives.

## 5.1 Baseline Experiments

We perform experiments using two baseline architectures: the U-Net architecture used in the construction of our dataset and the DeepLabV3+ architecture [8] which achieved a state-of-the-art result on a task similar to ours: pixel level land use classification with 5 classes [36]. We used the same architecture as [8] with an Xception [35] backbone, and trained the model from scratch (details in Supplementary A.2.3). In all experiments below, we further balance the training data by filtering out images with 30% or more background pixels, and weigh each class's contribution to the loss function to ensure that classes in the minority affect the loss as much as those in the majority. We use [49]'s custom weighting function where each class's weight is calculated as $\frac{1}{\log(1.02+\frac{N_c}{N})}$ where $N_c$ is the number of pixels of class $C$ and $N$ is the number of total pixels.

In our first experiment, we train both models on the training set and report results on our test set on Table 6. We also show confusion matrices in Supplementary A.2.5. Table 6 shows that both models result in $\sim 90\%$ or higher precision for all classes except for the nonresidential class (80.9% for U-Net and 86.8% for DeepLabV3+ at 50% IOU). Further inspecting the confusion matrices, when these neighborhoods are misclassified, it is usually to the wealthy or background classes. This could be because many industrial zones and wealthy neighborhoods are surrounded by non built-up land.

| Provinces | U-Net [54] | | | DeepLabV3+ [8] | | |
|---|---|---|---|---|---|---|
| | mIOU | Accuracy | Cohen's Kappa | mIOU | Accuracy | Cohen's Kappa |
| Gauteng | 0.6838 | 0.8153 | 0.7537 | 0.6294 | 0.7766 | 0.7021 |
| Limpopo | 0.7085 | 0.83 | 0.7733 | 0.6520 | 0.7900 | 0.7200 |
| Mpumalanga | 0.7074 | 0.8310 | 0.7746 | 0.6499 | 0.7907 | 0.7210 |
| Kwa Zulu-Natal | 0.7066 | 0.8280 | 0.7706 | 0.6605 | 0.7961 | 0.7281 |
| Free State | 0.7029 | 0.8257 | 0.7676 | 0.6055 | 0.7571 | 0.6761 |
| North West | 0.7592 | 0.8610 | 0.8147 | 0.6969 | 0.8211 | 0.7615 |
| Northern Cape | 0.6859 | 0.8148 | 0.7531 | 0.6289 | 0.7744 | 0.69915 |
| Eastern Cape | 0.6950 | 0.8207 | 0.7601 | 0.6371 | 0.7790 | 0.7054 |
| Western Cape | 0.7517 | 0.8571 | 0.8094 | 0.6958 | 0.8189 | 0.7585 |

Table 7: Mean Intersection over Union (mIOU), Accuracy and Cohen's Kappa after training U-Net [54] and DeepLabV3+ [8] models on 8 prvoinces and testing on the $9^{th}$ unseen province.

Our second experiment seeks to understand whether data from other provinces can be used to train models classifying land use in unseen provinces. To this end, we perform 9 experiments, each of which uses images from 8 provinces as training and validation data (6 for training and 2 for validation) and classifies pixels on images in the $9^{th}$ province. Table 7 shows the results for all provinces and confusion matrices are in the supplementary materials (A.2.4). The U-Net and DeepLabV3+ based models were able to classify pixels with an average Cohen's Kappa value of 0.77 and 0.70 respectively, and average mean IOUs of 0.7 and 0.6 respectively.

Table 8 shows the precision for each class and province at 50% IOU. That is, we measure the precision for each class, taking only segmentations of neighborhoods achieving an IOU of 50% or more in each image as true positives.

We can see that both models achieve a precision of $\sim 90\%$ or more, with almost perfect precision for the background class in spite of its prevalence. We attribute this to balancing and weighting our

| Provinces | U-Net [54] | | | | DeepLabV3+ [8] | | | |
|---|---|---|---|---|---|---|---|---|
| | W | NW | NR | B | W | NW | NR | B |
| Gauteng | 0.8971 | 0.9340 | 0.9070 | 0.9880 | 0.8995 | 0.9203 | 0.8912 | 0.9912 |
| Limpopo | 0.8981 | 0.9241 | 0.9077 | 0.9868 | 0.8997 | 0.9140 | 0.9024 | 0.9900 |
| Mpumalanga | 0.8993 | 0.9264 | 0.9089 | 0.9867 | 0.9024 | 0.9221 | 0.9048 | 0.9899 |
| Kwa Zulu-Natal | 0.8990 | 0.9216 | 0.9046 | 0.9859 | 0.9028 | 0.9140 | 0.8991 | 0.9894 |
| Free State | 0.8952 | 0.9254 | 0.9093 | 0.9881 | 0.8960 | 0.9059 | 0.8863 | 0.9910 |
| North West | 0.8967 | 0.9315 | 0.9117 | 0.9872 | 0.8953 | 0.9288 | 0.9074 | 0.9903 |
| Northern Cape | 0.8971 | 0.9294 | 0.9097 | 0.9872 | 0.8985 | 0.9263 | 0.9052 | 0.9901 |
| Eastern Cape | 0.9063 | 0.9189 | 0.9000 | 0.9863 | 0.9077 | 0.8965 | 0.8850 | 0.9896 |
| Western Cape | 0.8973 | 0.9319 | 0.9128 | 0.9878 | 0.8944 | 0.9293 | 0.9085 | 0.9908 |

Table 8: Precision at 50% IOU per province per class. W is the Wealthy class, NW the Non-Wealthy class, NR is Nonresidential and B is Background.

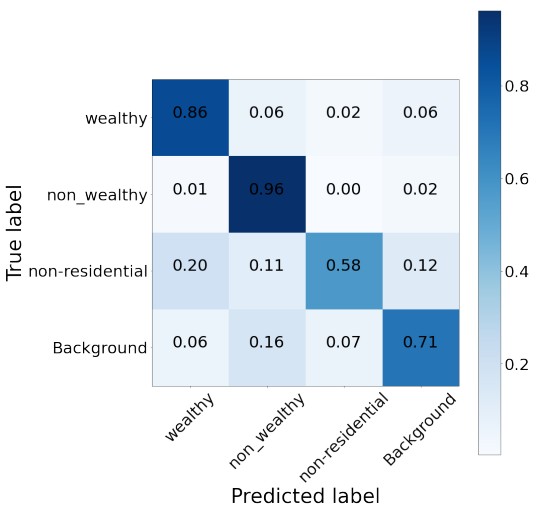

Figure 5: Confusion matrix for the U-Net model tested on Gauteng and trained on the rest of the provinces. See more confusion matrices in supplementary materials.

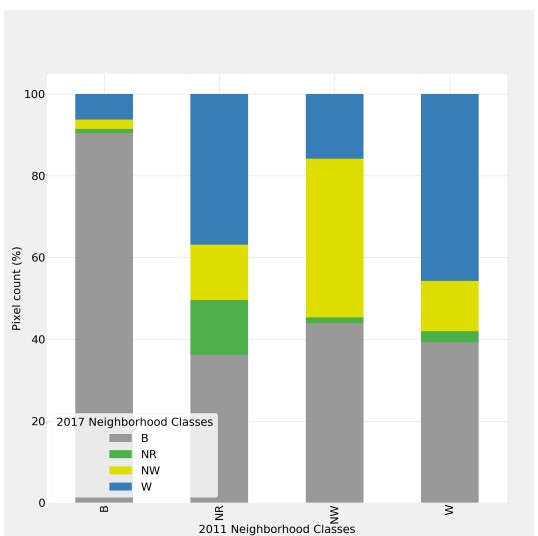

Figure 6: Estimated percentage of pixels that changed to other classes between 2011 and 2017 for each class.

training data. The precision for non wealthy neighborhoods is also over 90% for all provinces. Non wealthy neighborhoods consist of townships, collective living quarters, informal settlements and villages, where the latter two can be difficult to identify using satellite imagery due to their small sizes and irregular patterns (see Supplementary A.2.6). Governments may also not update datasets frequently enough to include new dwellings in sparsely populated settlements that can be quickly created and demolished throughout the country. Figure 5 shows that nonresidential neighborhoods in Gauteng are once again classified with much lower accuracy (58%) than other classes. Similar to our first experiment, these neighborhoods are often misclassified to the wealthy or background classes.

## 5.2 Studying the Evolution of Neighborhoods in South Africa

Here, we give examples of preliminary results studying the evolution of neighborhoods in South Africa, a task which we plan to perform in more detail in the future, to illustrate how our dataset can be used for this task. We ask: How have neighborhoods changed in Gauteng, one of the most populous provinces in South Africa? Have the sizes of townships increased or decreased on average? Have the number of wealthy neighborhoods increased or decreased?

**Methodology.** To answer this question, we used our U-Net model described in section 4 (trained on 2011 images and labels from Gauteng), and performed our inference on images from 2017. Note that since there are no labels for 2017, we cannot quantify the accuracy with which we perform this task. We hope to do a detailed error analysis in the future and look for additional data sources to help measure our accuracy. Here, we corroborate some trends we find through other studies [27, 22, 21, 29].

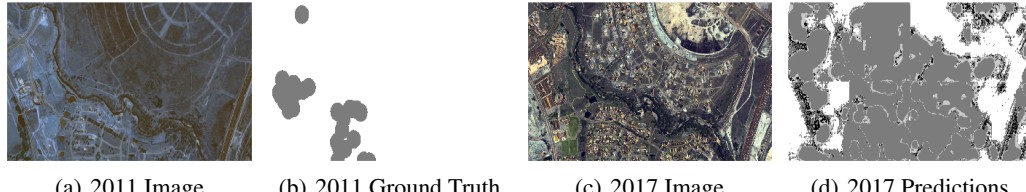

(a) 2011 Image      (b) 2011 Ground Truth      (c) 2017 Image      (d) 2017 Predictions

Figure 7: Examples of the change detected between 2011 and 2017 images in a wealthy neighborhood near a big mall. Dark gray: Wealthy Neighborhood, White: Background.

We first downsampled the 2017 images to the same resolution as those from 2011 $(21, 760 \times 21, 760)$, and then followed the same procedure as Section 4 to further process the images. To quantify the amount of change between the number of pixels belonging to each neighborhood in 2017 vs. 2011, we went through the following steps. (See supplementary materials A.2.7 for more detail).

1. Convert RGB masks into grayscale.

2. Store the difference in grayscale values between the 2017 predicted masks and 2011 ground truth masks. This allows us to see the "raw" difference in image masks, where no change would have the value zero and each other change would have a unique grayscale value. For example, the change from background to wealthy neighborhood is represented by 131.

3. Blur the difference image using a Gaussian kernel and threshold the output to reduce noise.

4. Store cluster centroids and the area of the cluster associated with each type of difference.

After these steps, we can reference each cluster centroid with the EA dataset (described in Section 3.2) and associate the changes we observe with specific municipalities and districts from 2011.

**Results.** Figure 6 shows the estimated changes associated with each class. For instance, we estimate that $\sim 40\%$ of neighborhoods remained the same type (e.g. $\sim 45\%$ of wealthy pixels are still wealthy). Our results also estimate that $\sim 90\%$ of vacant land stays vacant, but that $\sim 60\%$ of the vacant land that has been developed has been converted to wealthy residential neighborhoods, with $\sim 30\%$ developed to non wealthy residential neighborhoods. As noted in [27], Gauteng has seen a population explosion due to migration in addition to other factors, and construction has been dominated by wealthy formal residential buildings (Supplementary A.2.8). What is equally notable, according to the study, is the spread of shopping malls (up by 106% between 2001–2016), often surrounded by wealthy neighborhoods. Our estimates show similar trends, with Figure 7 showing an example of a wealthy neighborhood that was constructed around a mall between 2011 and 2017. While we are excited by our preliminary results, there are many sources of potential errors (Supplementary A.2.9) starting with the difference in image resolutions which we plan to investigate in more detail.

## 6    Broader Impact and Conclusion

We have introduced the first visual dataset of South Africa which can be used to analyze the effects of spatial apartheid, and described our iterative data annotation process that allowed us to assemble this dataset. We performed analyses to understand the characteristics of our dataset, and show the types of tasks that can be performed using it. We hope to enable those interested in studying and reversing the effects of spatial apartheid, to use this dataset. Coupling our analysis with census data could give further information on how the demographic makeup of the neighborhoods has changed, and working with policymakers could help us advocate for policies that desegregate neighborhoods.

As mentioned in Section 1, datasets such as this one could be aggregated with other datasets and used for applications which are not endorsed by us. This includes insurance companies using imagery to set high insurance rates for marginalized communities [3], financial organizations setting loans based on observations using satellite imagery [34], and entities associated with law enforcement and the military using satellite and associated imagery for drone and other types of targeting [57, 33]. Given this, we make our dataset accessible only for noncommercial use and through a request form which includes questions about intended use, the details of which can be found on the associated datasheet for the dataset.

## Acknowledgments and Disclosure of Funding

Thank you to Alex Hanna, Ben Packer, David Everatt, Danielle Wood, Memory Mhembere, Samy Katumba, Neil Lawrence, Xiao Wei for insightful discussions and suggestions. Funding for this project was provided through five sources: The South African Department of Science and Technology and the Council of Scientific & Industrial Research (as a masters scholarship award to Raesetje Sefala), Google (as a research award and compute credits to Raesetje Sefala), the Deep learning Indaba and Nvidia (Nvidia Titan V prize for best poster presentation at the 2018 Deep Learning Indaba summer school), and the Distributed AI Research Institute (DAIR).

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
