# OpenReview forum: "Constructing a Visual Dataset to Study the Effects of Spatial Apartheid in South Africa"
_NeurIPS.cc/2021/Track/Datasets_and_Benchmarks/Round2 — NeurIPS 2021 Datasets and Benchmarks Track (Round 2)_

### Official Review · Reviewer_Fs1R · 2021-09-02
**It is a human geography thematic dataset and is not suitable for machine learning or computer vision.**

**Rating:** 4
**Confidence:** 4
**Correctness:** The process of the dataset constructi…
**Clarity:** Please see 5,6 in Weaknesses

**Strengths:**

1. It is the first publicly available dataset to study the evolution of spatial apartheid.
2. The publicity of the dataset is helpful to the field of human geography, which may promote the relevant research of studying the evolution of neighborhoods in South Africa.

**Weaknesses:**

1. The contributions are not very clear and are supposed to be highlighted in the Abstract.
2. Some novel insights require to be added referring to the experiments.
3. More advanced remote sensing semantic segmentation methods need to be extended.
4. Article organization needs adjustment. The descriptions of data preparation can be reduced or moved to the Appendix.
More statistical feature analysis needs to be clarified to enhance the value of this dataset.
5. The data generation process includes buffer and topology analysis. These parameter-sensitive operations will bring some noise. How to control the accuracy and quality of each step?
6. Why are the evaluation indicators in Table 7 and Table 8 different？ The $mAP_{50}$ needs to be clarified because this is an object detection metric. Are there any instances in the labels?


**Additional Feedback:**

Please see Weaknesses

**Documentation:**

The appendix helps for better understanding the details and documenting the process.

**Relation To Prior Work:**

Yes. It is the first publicly available dataset to study the evolution of spatial apartheid.

**Summary And Contributions:**

The dataset is designed for studying the evolution of spatial apartheid in South Africa.
This dataset contains 6,768 remote sensing images, covering 9 provinces for 4 classes of neighborhoods.
Several analyses have been performed to understand the characteristics and changes in South Africa.
However, this dataset seems to be designed for human geography thematic tasks, which may not be suitable for NeurIPS.
Besides, the benchmarked experiments are relatively weak and can not inspire future model development in machine learning.

---

> ### Author Response · Authors · 2021-09-29
> **Responding to feedback and questions from reviewer**
>
> We thank you for your review and address your feedback below. Added text in our revised submission is highlighted in yellow, and references noted below are in a pdf titled References for Reviewer 3 with our supplementary materials.
>
> 	1.	dataset…may not be suitable for NeurIPS.
> Our dataset and methodology are based on semantic segmentation, keeping real world impact in mind. We argue there should be more works in ML tied to real world problems, especially at this NeurIPS track created to address the lack of venues for critical data work (https://neurips.cc/Conferences/2021/CallForDatasetsBenchmarks).
>
> 	2.	benchmarked experiments…can not inspire future model development in ML.
> Our dataset was created to analyze spatial apartheid using ML techniques. During data collection and experimentation, we used state of the art semantic segmentation methodology as baselines [5,8] and showed various failure cases (Section 5, Supplementary A.2.4–A.2.9) with a lot of room for improvement. Thus, our dataset can be used to study architectures that close this gap. Some ML works that can be advanced by our dataset are:
>
> 	•Diverse datasets to train and evaluate semantic segmentation models for land use classification: Biased datasets result in models that only work in certain geographies [10-13]. Our dataset is the only land-use dataset covering an entire African country.
>
> 	•Development of models for highly imbalanced classes, that better generalize to different image resolutions, and images taken at different times.
>
> 	•Iterative dataset construction and stewardship methodologies under limited resources [14].
>
> 	•Interdisciplinary research in ML: Our dataset gives more accurate information than proxies such as nightlights from satellite images [15-19] used to study group level socio economic factors.
>
> 	3.	Contributions are not very clear…need novel insights
> Our contributions are listed in the abstract and expanded on in the introduction.  We gather a novel dataset using an iterative data annotation methodology, with resource constraints (details in Section 4 and Section A.1). In Sections 5 and A.2, we provide insights into results, analyses, challenges, and new results studying South African neighborhoods. Without specific feedback, it is unclear what additional novel insights the reviewer is looking for, and we fail to see the lack of clarity of our contributions.
>
> 	4.	More advanced remote sensing semantic segmentation methods need to be extended.
> We show baseline results on two state-of-the-art models used in similar tasks to ours [5,6].  While there are data preprocessing methods that may improve results (e.g. [21,22]) and additional research to be done here, we use baseline models with little pre-processing to understand our dataset as is done by similar works introducing new datasets (e.g. [7-9,20]). If there are specific models the reviewer hoped to see, we’d like to understand which ones and why.
>
> 	5.	descriptions of data preparation can be reduced or moved to the Appendix.
> As this paper is primarily focused on introducing a new dataset with the methodology used to create it, we disagree that descriptions of data preparation should be relegated to the appendix. We have also added more detail to Section 3 as requested by Reviewer 2.
>
> 	6.	More statistical feature analysis needs to be clarified...
> We show the distribution of pixels per class (Tables 4,6 supplementary Table 2), which is what is reported by prior works introducing similar datasets (e.g. [8,22,26]), provide maps (e.g. supplementary Figures 1 and 9), and many samples of the dataset and experimental results (e.g. Section A.1.5). We’d like to understand what specific additions the reviewer is recommending and why.
>
> 	7.	How to control the accuracy and quality of each step?
> Thank you for pointing this out. We've added potential errors associated with different components in Section 3, and a discussion in the answer to datasheet Q13, in addition to those in Sections A.1.3, A.1.4 and A.2.6. E.g. there are labeling errors related to the building count dataset that can only be verified and corrected through visiting the locations, and errors related to labeling townships in 2021 for images in 2011. As noted by Reviewer 2, the buffer adds noise used to protect residents’ privacy. This injection of noise into datasets is a well known technique (e.g. in differential privacy [27]). In section A.1.4, we discuss how we chose our buffer size and its impact on the accuracy of our annotations.
>
> 	8.	Why are the evaluation indicators in Table 7 and Table 8 different？The mAP50 needs to be clarified…
> Table 7 and 8 show different insights for the same experiment, with Table 8 listing the precision at 50% IOU per province per class. mAP50 in this case is a typo which should have been Precision @ 50% IOU as we are not averaging over classes. We thank you for pointing this out. We have fixed this typo in the paper and added clarifying text in Section 5.1.

---

> > ### Comment · Reviewer_Fs1R · 2021-09-29
> > **Response to the authors**
> >
> > Thanks for your explanations.
> >
> > I still think the current format of this paper is not suitable for NeurIPS.
> >
> > - The construction of this dataset is automatically generated by a series of topology analyses.
> > Not only it is difficult to control errors but the generated mask seems strange in Figure 4.
> > There will be lots of errors at the edges, which poses a big hidden danger to the calculation area.
> >
> >
> > - The motivation is good but what's the difference between this dataset and other land-cover segmentation datasets?
> > What kind of segmentation methods can be inspired by the unique characteristics of this dataset?
> > Simple UNet  and DeepLabV3 are difficult to reflect the current bottleneck problems because many SOTA segmentation models (HRNet[1], OCNet[2], RSNet[3], S-RA-FCN [4], etc.) have pushed the community further. At least few remote sensing segmentation methods are needed to be disscussed and evaluated.
> >
> >
> > [1] Wang J, Sun K, Cheng T, et al. Deep high-resolution representation learning for visual recognition[J]. IEEE transactions on pattern analysis and machine intelligence, 2020.
> >
> > [2] Yuan Y, Huang L, Guo J, et al. Ocnet: Object context network for scene parsing[J]. arXiv preprint arXiv:1809.00916, 2018.
> >
> > [3] Wang J, Zhong Y, Zheng Z, et al. RSNet: The search for remote sensing deep neural networks in recognition tasks[J]. IEEE Transactions on Geoscience and Remote Sensing, 2020, 59(3): 2520-2534.
> >
> > [4] Mou L, Hua Y, Zhu X X. A relation-augmented fully convolutional network for semantic segmentation in aerial scenes[C]//Proceedings of the IEEE/CVF Conference on Computer Vision and Pattern Recognition. 2019: 12416-12425.
> >
> > - The authors also claim that "Development of models for highly imbalanced classes, that better generalize to different image resolutions, and images taken at different times.  " The corresponding exploration experiments (multi-resolution or multi-temporal fusion) need to be added at least to show the characteristics of the dataset.
> >
> > - The metric is strange, IOU of 50% is not the mainstream evaluation standard for semantic segmentation.
> > The readership of NeurIPS prefers to use IoU for evaluation to facilitate comparative analysis.

---

> > > ### Author Response · Authors · 2021-09-30
> > > **We answer your questions below (noted references are in a pdf “New References for Reviewer 3” in supplementary materials).**
> > >
> > >
> > > 	1. difficult to control errors…generated mask seems strange…lots of errors at edges…
> > > Additional sample images/masks are in Supplementary figures 4,7,8, 22, and datasheet Q5. The circular nature of some masks is to be expected given our buffing (A.1.4). Each 256X256 image covers 2.56kmx2.56km as 1 pixel covers 10 meters–a small area relative to neighborhoods. Yes, potential error at edges due to noise injection is to be expected, similar to the tolerance in differential privacy for the same reason. However, those errors are very small for our task: a potential classification error covering max. 77.7m^2 doesn’t affect our ability to accurately classify neighborhoods as we’re not interested in segmenting specific objects like trees or cars. These are the types of tradeoffs made while creating a dataset and determining tasks it can be used for.
> > >
> > > 	2. What's the difference between this dataset and others…?
> > > Table 4 in [2] shows a few public datasets for remote sensing. As discussed in Section 2, our dataset covers different tasks from many others, and many don’t have diverse labels (e.g. don’t distinguish between neighborhood types in “urban” class). Its one of the few datasets covering an entire country, and the only one covering an African one (consecutive tiles and labels for ~1.2 million km^2). In addition to providing geographic diversity [10-13], the scale opens new challenges and applications like ones we study. To create a dataset like this with limited resources, we innovated in dataset creation methodology, and our task of interest requires innovations in model development.
> > >
> > > 	3. What kind of segmentation methods can be inspired…?
> > >
> > > There are many failure cases/challenges discussed in Section 5 and throughout the supplementary materials. Further points of inquiry can be:
> > >
> > > 	•Model development for datasets where 90% of data belongs to 1 class but distinguishing between classes covering 10% of the data is also crucial.
> > >
> > > 	•What metrics are most important in this task? Working on segmentation models in the abstract without considering specific goals doesn’t allow this type of study, which is crucial as the disconnect between ML development and real world applications is leading to harmful deployments [14].
> > >
> > > 	•The dataset is large, covering an entire country. How can we create more efficient segmentation models?
> > >
> > > 	•Dataset development methodology is a crucial but ignored aspect of ML. How can we innovate on it under resource constraints?
> > >
> > > 	•Studying properties of different models. Why does a simple UNet outperform DeepLabV3+ when the opposite is true in datasets like [3]? What aspects of these architectures determines types of datasets they’re applicable on? At which scale does a pre-trained backbone hurt vs. help?
> > >
> > > 	•Our experiments in Section 5.2 are done by applying models to images taken at a different time and with a different resolution. We expect to do much more work in this area using our dataset as this is crucial for our end goal.
> > >
> > > 	4. Simple UNet and DeepLabV3 are difficult to reflect the current bottleneck problems…
> > > We showed two models commonly used in remote sensing applications achieving state of the art results in tasks comparable to ours. Variations of these models, and many others, can be experimented on using our dataset, which is why we are introducing it. [4] applies HRNet to road extraction (a different task from ours). We found [5] on HRNet for remote sensing, published on August 5 2021, 22 days before our paper was submitted. It shows <0.2% difference between U-Net, DeepLabV3+ and their best HRNet model on land cover classification tasks (still different from ours), consisting of cars, trees etc. OCNet was published on May 24, 2021–and not applied to remote sensing but other tasks. S-RA-FCN from CVPR 2020 notes “However, our understanding of how these relation modules work for segmentation problems is preliminary and left as future works.” The evaluation was on object centric datasets from areal images. They don’t compare with UNet (nor mention it in related works) or DeepLabV3+, the most commonly used models in land use classification. Still, the difference between their method and the simplest DeepLab model (which can have 4%+ difference with DeepLabV3+), was <4%. Analyzing whether or not these modules add value in a dataset such as ours, is an interesting future work to look into. Thus researchers in ML, CV and other fields can use our dataset to investigate different types of questions.
> > >
> > > 	5. The metric is strange…not the mainstream
> > > We report all of these metrics, including IOU, accuracy, and additional ones such as Cohen’s Kappa which  gives insights into imbalanced datasets. We did not omit IOU (Table 7) but show precision at different IOUs in addition (Table 8): also commonly done in similar tasks, e.g. [3,8]. Its important to present different metrics showing different insights depending on the end goal.

---

### Official Review · Reviewer_a5QB · 2021-09-16
**20210916 Review of the "Spatial Apartheid" dataset (Sefala et al, 2021)**

**Rating:** 8
**Confidence:** 3

**Strengths:**

The strengths of this paper are 3-fold, and are highlighted here. (Please refer subsequent sections for further details).

* It is built using a collection of open data - per the `Datasheet` PDF: South African National Space Agency satellite imagery + Eskom  building count dataset + EA dataset (the latter is public South African census data).
* The preprocessing workflow is adaptable to continually updated satellite images -- per e.g. Section 5.2, Supplementary Figure 23 -- which will appeal to inter- and cross-disciplinary studies as highlighted in the Summary And Contributions above. To elaborate, as the data is provided in image format, it is easily processed by researchers in other disciplines merely using standard bitmap image processing techniques (as the authors have provided an example in Section 5.2 - the mere difference/delta in pixel intensity over time can detect changes in population density patterns).
* The workflow is describe in a clear, concise manner; and the authors have provided a thorough examination of use cases and (most importantly) ethical and societal issues in the supporting material.

**Weaknesses:**

There are several issues which may need clarification, mainly the process of assembling and preprocessing the dataset.
These are intended to improve the quality of the dataset and the documentation, should the dataset be recommended for acceptance.

* Meanings of colour maps - In Section 3, in the course of documenting the data processing steps, it would be preferable if the use of colour was standardised throughout, and if possible, a colour-blind/colour vision deficiency-friendly palette was employed.
For example, when following the processing pipeline in Section 3 onwards, Figure 2 (EA polygons) indicate *farms* in a shade of cyan/turquoise. In Figure 3(b)(top), *farms* are a light purple, while in Figure 3(b)(bottom)/3(c)(left), and `Datasheet` (point #4) *farms* are in a shade of bright green, and finally for Figure 3(c)(right), it goes back to a brighter/pastel shade of purple. This makes the section a bit hard to follow (unless the reader has, beforehand, studied the Supplementary Material to identify the trends of colour palettes for different stages).

* Combining the sources of mask and map data - In Section 3, satellite images from SPOT are restricted to the year 2011, to correspond with the year the Enumeration Areas (EAs) dataset was created in 2011. However, the "Geo-referenced buildings ... shapefile dataset was created by Eskom ... and consists of building count data in South Africa from 2006 to 2016". This is unclear. Further, the Wikipedia-based disambiguation process will need to be elaborated. (will be expanded in Correctness below).

**Additional Feedback:**

I commend the authors for a strong submission to NeurIPS 2021 Track Datasets and Benchmarks; where the empirical data can greatly benefit researchers from various disciplines in studying contemporary social issues.

However, there are some issues above which will need to be resolved or clarified before it is ready for acceptance: the issues raised above are mainly for completeness (in terms of methodology), especially in terms of how the dataset is compiled.

I wish the authors all the best for NeurIPS 2021.

**Clarity:**

The paper is very well written. No major issues or typographical errors found.
Some ideas on improving the visualisation of the dataset (including intermediate steps) are discussed in the Correctness section above.

Kudos to the authors for a clear and thorough documentation in the Supplementary Material to anticipate questions that reviewers/dataset users might have.

**Correctness:**

The overall construction of the dataset is performed in a sound way.
* The Supplementary Material section has done a good job at explaining the entire pipeline, alongside a generous collection of images to illustrate the dataset in practical use.
* Figure 9 was especially useful in visualising the train/test sample split for the neighbourhood classiﬁcation model.
* The authors have also taken great care in ensuring the geo data projections are standardised, and ensuring that the classes are balanced (Section 4).
Hence, overall, the processing pipeline for constructing, validating, and testing the dataset appears sound.

Two issues in the 'source data' can be addressed further.
One of the key items that the reviewer would like the authors to address is the use of the "Geo-referenced buildings dataset" (hereinafter, the "shapefile" dataset) - as discussed in the main manuscript, Section 4, "consists of building count data in South Africa from 2006 to 2016". However, it is not clear that the dataset is
(a) built in the ten year time period, whereby it runs the risk of having buildings > 2011 *not reflected* in the choice of EA + SPOT images; or
(b) similar in versioning as the SPOT dataset, i.e. one per year, and that the 2011 version was chosen to align with the latest possible EA year.
The `Datasheet` file, under Collection Process->point 18, mentions further details about the shapefile dataset including a link to a technical report. It would be optimal if the authors can cite this report in the main manuscript, and provide some extra details about the specifics of the shapefile construction (strengths/weaknesses/considerations), as Section 3 currently lacks details (when compared to, say, the EA dataset and the SPOT dataset).

Another issue is also in Section 3: "One shortcoming of the EA dataset is that townships are grouped with suburbs... which does not allow us to distinguish between them... each EA is associated with a name... looked up a list of South African townships on Wikipedia and labelled the EAs". The use of Wikipedia to label the EAs is one way to get past the limitation; however more information needs to be provided on this process - e.g. how many total EAs require Wikipedia disambiguation, with examples of the disambiguation process? Further for the recruitment of "10 volunteer students who grew up in townships to aid in labelling townships" (per `Datasheet`), some comments on the methodology will be preferable (e.g. each of the X ambiguous townships was labelled by Y students by consensus/majority vote)?

**Documentation:**

The `Datasheet` PDF is very thorough and well-written. It provides all the details for data collection and organization, availability and maintenance, and ethical and responsible use. Hosting is accomplished via the Google Cloud Platform.


**Ethics:**

There are no ethical concerns warranting review.

The authors have done an excellent job outlining the data governance plans, limiting the use of the dataset to avoid unethical use cases, as well as, in general, considered most (if not all) ethical concerns in their documentation.

Further, using a buffer with a roughly 77.7m^2 radius (0.0007deg) provides some 'noise' to prevent overly-accurate pinpointing of building coordinates, with serves to protect building residents' privacy.

**Relation To Prior Work:**

Yes, how this work differs from previous contributions is clearly discussed in Section 2.

**Summary And Contributions:**

To briefly summarise, the submission is a dataset "to study the evolution of spatial apartheid", in the form of "geo-referenced satellite images covering the entire country of South Africa and a corresponding mask of neighborhoods labelled according to their type". As this dataset covers the entirety of South Africa, the submission provides a wide coverage, while at the same time provides a high-enough resolution for the research community.

The contribution is strong, built using relatively open data sources, and has the potential to contribute to meaningful cross-disciplinary research, beyond ML --  from geographic studies, to social science, to economics and public policy, etc. -- due to its potential ease of use, and above all, thoroughness.

---

> ### Author Response · Authors · 2021-09-29
> **Incorporating suggestions from reviewer**
>
>
> Answer: We thank you for your incredibly detailed review and suggestions. We briefly address some of your questions and suggestions here.
>
> 	1.	Meanings of colour maps….
> Thank you for this suggestion. We agree that we should standardize our colour maps and that we should utilize a colour-blind/colour vision deficiency-friendly palette. We are revising all images consulting with this website https://www.tableau.com/about/blog/examining-data-viz-rules-dont-use-red-green-together in order to make our visulaizations colouf-blind friendly and consistent. We will have the revised figures on the camera ready version of our paper.
>
> 	2.	Clarification of use of building count data in South Africa from 2006 to 2016 and adding details to Section 3
> Thank you for pointing this out. The building count dataset is indeed per year. That is, there is building count data for 2006, 2007, 2008 up to 2016. To assemble our labeled dataset, we used images and building count data from 2011 because the EA data was only available for 2011. Thus, we provide labels for the 2011 satellite images, to train and evaluate models, and provide unlabeled satellite images for the other years between 2006-2017 for further analysis. We have text in section 3 to clarify this point (highlighted in our revised submission). An excerpt of the text is below but we have truncated it due to space limitations. You can find the full text in our revised submission.
> “The dataset captures geographical coordinates of formal, informal and non-dwelling structures per year over a period of 10 years…To annotate our dataset, we use the building count data from 2011 consisting of 12,515,847 buildings in South Africa in 2011, as the EA dataset is only available in 2011.
> The only publicly available reports we have been able to find for the dataset are from 2007 and 2010, which describe the labeling procedure and potential sources of error [12, 44]. As noted in [44],  “All the mapping and classification of the structures are done through image interpretation and no field work has been conducted at this stage of the project”
> The initial dataset was created in 2006, and used as a basis for 2007, only updating buildings that changed in the last year (either new buildings or demolished buildings). Data for subsequent years was also created using the same procedure: using the dataset from the previous year as the base layer for the next year of interest. A random sample of the dataset was selected from across the country, and evaluated by independent labelers which corrected points with high false positives and false negatives…”
>
> 	3.	Clarification of Wikipedia-based disambiguation process.
> We agree that we should give further details on our Wikipedia-based disambiguation process. We have added the text to Section 3 describing our process (highlighted in our revised paper). An excerpt of the text is below but we have truncated it due to space limitations. You can find the full text in our revised submission.
> “The EA dataset consists of attributes representing data aggregated at different resolutions. EAs are the smallest unit of aggregation, followed by Sub-places, Main-places, Districts and continuing on to Provinces. In order to identify townships that may be annotated as formal residential areas, we took the following steps.
> We first obtained a list of all the Mainplaces that are annotated as formal residential areas, resulting in 1,655 Mainplaces consisting of a combination of suburbs and townships. Then, we recruited 10 graduate students at the University of Witwatersrand. Each student was born and raised in one of the 9 provinces, with 2 students from the KwaZulu-Natal province, the largest province in South Africa (by population size and area). The students’ task was to check if any of the Mainplaces that were labelled as formal residential areas were indeed townships. Each of the 10 students went through all 1,655 Mainplaces and labeled those they believed to be townships as such, along with their level of certainty (certain or uncertain). For this procedure students often asked their relatives or others living in the townships to verify their labels. If 3 or more students agreed that a Mainplace is a township, then we labelled that neighbourhood as a township. To validate the labels, we further used additional sources such as Wikipedia and property websites such as https://www.privateproperty.co.za/ and https://www.property24.com/ to retrieve a list of townships and suburbs in South Africa respectively. We assembled a list of 362 townships from Wikipedia….”
>
> References
>
> [12]Eskom.  2007 Spot Building Count Update Report. https://github.com/sefalab/Spatial_Project/blob/master/2007%20SPOT%20BUILDING%20COUNT%20UPDATE_report.pdf, 2007.
>
> [44]Nale   Mudau   and   ESI-GIS. SPOT   Building   Count   supports   informed   decisions. https://www.ee.co.za/wp-content/uploads/legacy/PositionIT%202009/PositionIT%202010/SPOT.pdf, 2010.

---

### Official Review · Reviewer_sHBj · 2021-09-18
**Visual Dataset to Study the Effects of Spatial Apartheid in South Africa**

**Rating:** 7
**Confidence:** 4
**Clarity:** The paper is well written.

**Strengths:**

+ The proposed task is interesting for ML researchers and results can be of interest to researchers of other areas. From this point of view, the expected impact of this dataset goes beyond the strict field of ML.
+ There are several land-use datasets that represent similar tasks, but this one corresponds to a poorly represented environment.
+ Ethical and social implications are clearly identified.


**Weaknesses:**

+ Access limitations, depending on the way they are applied, could severely limit the potential impact of the dataset. The access form is not available. It is necessary to define in clear terms which applications are legitimate and which are not.
+ Modern land-use datasets are richer, in terms of features, than the one presented in this paper because of the limitation of satellite images (resolution and spectral properties). It is not clear if the provided data is sufficient to develop better ML models to increase classification performance. On the other side, to study the effects of spatial apartheid, there is also a lack of additional data. Because of this, the presented dataset needs more data to maximize its potential impact.

**Additional Feedback:**

+ Complete related work with a broader survey of existing land-use datasets.
+ Consider the enrichment of the dataset with pixel level, open-source, geographical data.
+ Define in clear terms legitimate uses of this dataset.

**Correctness:**

+ The dataset creation methodology is well described and there are no concerns related to this point.
+ Evaluation methods are well selected.


**Documentation:**

+ There is sufficient detail on data collection and organization.
+ Information about availability and maintenance is too limited.
+ Ethical and responsible use should be better defined.

**Ethics:**

+ Ethical issues have been clearly identified.
+ Datasheets are used to document the dataset.

**Relation To Prior Work:**

The most related class of datasets are land-use datasets, but some of the most popular datasets are not considered:
+ U.S. Geological Survey. National Land Cover Database, New York.https://cugir.library. cornell.edu/catalog/cugir-009031, 2016.
+ Seyed  Majid  Azimi,  Corentin  Henry,  Lars  Sommer,  Arne  Schumann,  and  Eleonora  Vig. Skyscapes  fine-grained semantic understanding of aerial scenes.  In Proceedings of the IEEE/CVF International Conference on Computer Vision (ICCV), October 2019.
+ Emmanuel Maggiori, Yuliya Tarabalka, Guillaume Charpiat, and Pierre Alliez. Can semantic labeling methods generalize to any city? the Inria aerial image labeling benchmark. In IEEE International Geoscience and Remote Sensing Symposium (IGARSS). IEEE, 2017.
+ Adam Van Etten, Dave Lindenbaum, and Todd M Bacastow.  Spacenet:  A remote sensing dataset and challenge series.arXiv preprint arXiv:1807.01232, 2018.


**Summary And Contributions:**

This paper introduces the first publicly available dataset to study the evolution of spatial apartheid, using a large number of high-resolution satellite images of 95 provinces in South Africa.  It includes pixel-wise labels for 4 classes of neighborhoods.
The paper presents some baseline experiments with state-of-the-art models commonly used in land use segmentation tasks.

---

> ### Author Response · Authors · 2021-09-29
> **Addressing feedback and suggestions from Reviewer**
>
> We thank you for your detailed review and address some of your questions below. We also attach a pdf titled References for Reviewer 1 with a list of 22 noted references (in the same folder as our supplementary materials).
>
> 1. Related work: thanks for your suggestions. The datasets introduced in these 4 papers have different objectives to ours, described below, and we have added text in Section 2 discussing them (highlighted in our paper).
> Azimi et al. perform land-use classification for classes such as urban, agriculture, forest, etc without further granularity within the urban class. Maggiori et al. introduce a dataset annotating buildings, cars, roads, trees, sidewalks etc. in the US and Austria–a different objective from ours. Our goal is to classify neighborhood types, using clusters of buildings and other characteristics of each neighborhood. The last two papers introduce datasets to detect the location of individual buildings, with Etten et al. also annotating roads--a different task from ours as described above.
>
> 2. Enrichment of the dataset with pixel level, open-source, geographical data….
> We provide pixel level masks annotated by 12 neighborhood classes and the collapsed 4 neighborhoods, available in png (can be converted to the geotiff format with our provided code), and shapefile formats. To follow recommendations for responsible dataset use [1,2,3,4], we track dataset usage through an access form and a licence, noting acceptable and unacceptable use cases. Other neighborhood classification datasets we have seen similarly provide pixel annotations with the same formats as ours (e.g. [5-7]). Thus, we are wondering what specific information/sources the reviewer has in mind in order to enrich our dataset.
>
> 3. On the other side, to study the effects of spatial apartheid, there is also a lack of additional data…
> Experiments in section 5.2 show examples of questions that can be investigated using our dataset and computer vision techniques. Our future work will be partnering with geographers and urban studies experts like Samy Katumba who authored many works analyzing spatial apartheid (e.g. [8,9]). While we agree that additional data can supplement our work and aid in the analysis (such as census and survey data as noted in Section 6), we believe that our dataset, presenting pixel level level information available for datasets in other countries (e.g. [5,6,7]), is an important first step to analyze the evolution of South African neighborhoods.
>
> 4. It is not clear if the provided data is sufficient to develop better ML models…
> Our dataset was created for the task of analyzig spatial apartheid using computer vision techniques. Both in our data collection and experimentation processes, we used state of the art semantic segmentation methodology as baselines [10,11] and have shown various failure cases (Section 5, Supplementary A.2.4–A.2.9) which means there is a lot of room for improvement. Thus, our dataset can be used to experiment with architectures that can close this gap. A detailed study of model generalization and performance, and accurate semantic segmentation models, are crucial for our ability to perform our analysis of spatial apartheid. This is one of the reasons why we, a group of computer vision researchers, developed this dataset. Some works in ML that can be advanced by our dataset are:
>
> 	•Introducing diverse datasets to train and evaluate semantic segmentation models for land use classification: As shown by [14] and others, biased datasets result in models that only work in specific geographies [15,16,17], and our dataset is the only land-use dataset covering an entire African country.
>
> 	•To study the development of models for highly imbalanced classes, that better generalize to different image resolutions, and images taken at different times.
>
> 	•To study iterative dataset construction and stewardship methodologies under limited resources, and responsible data use [1].
>
> 	•To advance interdisciplinary research in ML and computational social sciences: Our dataset gives more accurate information than proxies such as nightlights from satellite images [18-22] used to study group level socio economic factors.
>
> 5. Access form and clear terms of use: The access form is located at https://forms.gle/6qV2ucQ6KUKYmL1J9, noted on the answer to Q42 in our datasheet. The URL should be accessible--please let us know if you are having trouble accessing it as that is not by design. The license is at https://github.com/sefalab/Spatial_Project/blob/master/licence.md, also linked in the access form. Our dataset can only be used for research purposes, and commercial or military use is prohibited. We also prohibit the use of trained models on applications listed in answers to Q39 of the datasheet, such as determining financial consequences for a group (e.g. interest rates, insurance prices or loans). We welcome any additional suggestions on clarifying dataset use and maintenance information.

---

> > ### Comment · Reviewer_sHBj · 2021-09-29
> > **Last revision**
> >
> > Thank you for your detailed comments. I mostly agree with all of them. I have updated my ranking accordingly.

---

### Decision · Program_Chairs · 2021-10-09

**Decision:**

Accept

**Comment:**

The data provided in the paper is of relevance for the NeurIPS data track. Based on reviewers opinions, and in particular after discussion with the authors, the paper achieves the minimum score required for publication at NeurIPS data track.